# Case-Based Off-Policy Evaluation Using Prototype Learning

**Anton Matsson**[1]                    **Fredrik D. Johansson**[1]

[1]Chalmers University of Technology

## Abstract

Importance sampling (IS) is often used to perform off-policy evaluation but it is prone to several issues—especially when the behavior policy is unknown and must be estimated from data. Significant differences between target and behavior policies can result in uncertain value estimates due to, for example, high variance. Standard practices such as inspecting IS weights may be insufficient to diagnose such problems and determine for which type of inputs the policies differ in suggested actions and resulting values. To address this, we propose estimating the behavior policy for IS using prototype learning. The learned prototypes provide a condensed summary of the input-action space, which allows for describing differences between policies and assessing the support for evaluating a certain target policy. In addition, we can describe a value estimate in terms of prototypes to understand which parts of the target policy have the most impact on the estimate. We find that this provides new insights in the examination of a learned policy for sepsis management. Moreover, we study the bias resulting from restricting models to use prototypes, how bias propagates to IS weights and estimated values and how this varies with history length.

## 1 INTRODUCTION

Historical data on decisions and outcomes provide opportunities for evaluating policies for future decision-making. For example, the prospect of using patient records to evaluate new policies for medication dosing in sepsis management has attracted recent attention [Komorowski et al., 2018, Gottesman et al., 2019]. An example of off-policy evaluation (OPE), this amounts to estimating the value of a target policy based on data gathered under a different so-called behavior policy; see e.g., Thomas [2015] for an overview.

Importance sampling (IS) methods [Precup, 2000] perform OPE by weighting observed outcomes by the density ratio of the target policy and the behavior policy. IS methods are often preferred over alternatives which rely on modeling outcomes or covariate transitions, due to their simplicity and the fact that behavior policies often are controllable or human made. Similarly, the equivalent strategy of inverse-propensity weighting is fundamental to the study of causal effects [Rosenbaum and Rubin, 1983, Hirano et al., 2003].

In practice, it is difficult to assess the quality of an IS value estimate. When the behavior policy is unknown and must be estimated from data, conditions which guarantee good estimates are hard to meet and rely on untestable assumptions [Rosenbaum et al., 2010, Namkoong et al., 2020]. Standard practices of inspecting weights [Li et al., 2019] and removing outliers [Crump et al., 2009] give only aggregate or per-sample perspectives on potential issues and are often insufficient for domain experts to reason about the validity of the result. There is a clear need to better inspect and diagnose importance sampling estimates.

In this paper, we propose estimating the unknown behavior policy using prototype learning [Li et al., 2018b, Ming et al., 2019]. The learned prototypes are selected cases from the input data, readily interpretable by a domain expert and representative of the input-action space. In healthcare applications, the prototypes are trajectories of former patients, and a prototype-based estimate of the behavior policy is analogous to how physicians use experience from previous patients to treat new ones. While offering transparency, a prototype model is flexible enough to model behavior policies in large and/or sequential input spaces.

Our main contribution is to use learned prototypes as an OPE diagnostic tool. In addition to enabling interpretation of individual predictions, we show that (a) prototypes can be used to describe areas of similarities/dissimilarities between behavior and target policies; and (b) prototypes induce a soft clustering which can be used to explain differences in value

*Accepted for the 38th Conference on Uncertainty in Artificial Intelligence* (UAI 2022).

for different policies. We elaborate on this idea in Section 3.3 and demonstrate our method in Section 4.1 using an example of sepsis management. Further, we study the added bias of restricting the model class to use prototypes and how this bias propagates to the IS weights in Section 4.2.

## 2 OFF-POLICY EVALUATION

Policy evaluation refers to estimating the *value* $V(\pi)$ of a *target policy* $\pi \in \Pi$, as defined below. We focus on the sequential case, where a policy is used to select an *action* $A \in \mathcal{A} = \{1, \ldots, k\}$ after a *history* $H \in \mathcal{H}$, comprising a sequence of previous actions and *contexts* $X \in \mathcal{X}$. The history until time $t$ is defined as $H_t := (X_0, A_0, X_1, A_1, \ldots, X_t)$, with $H_0 = X_0$. A policy $\pi : \mathcal{H} \to \Delta_{\mathcal{A}}$ is a map from a history to a distribution over $\mathcal{A}$. In a medical example, a context $X$ could correspond to information about a patient's state, an action $A$ to a medical intervention, and the target policy $\pi$ to new clinical guidelines.

The value of a policy $\pi$ is defined as the expectation of a *reward* or *outcome* $R \in \mathbb{R}$, accumulated after acting according to $\pi$. Here, we study the special case where a single reward is awarded at the end of the sequence, $R = R_T$, but our results generalize to the case where rewards are given after every action. Under the distribution $p_\pi(X_0, A_0, \ldots, X_T, A_T, R) = p_\pi(H_T, A_T, R)$, induced by the policy $\pi$, the value is $V(\pi) := \mathbb{E}_\pi[R]$.

Estimating $V(\pi)$ is trivial given a large enough number of samples from the target policy $p_\pi$. In off-policy evaluation (OPE), we have access to no such samples, but must estimate $V(\pi)$ using an observational dataset of $m$ samples $\mathcal{D} = ((h_{t_1}^1, a_{t_1}^1, r^1), \ldots, (h_{t_m}^m, a_{t_m}^m, r^m))$, drawn according to a distribution $p_\mu(H_T, A_T, R)$, controlled by a *behavior policy* $\mu \in \Pi$. In the medical example, the behavior policy represents current clinical practice. In this work, the behavior policy $\mu$ is unknown and an estimate $\hat{p}_\mu(A \mid H)$ is learned from the samples $\mathcal{D}$.

A common method for OPE is *importance sampling* (IS).[1] The IS estimator uses an estimate $\hat{p}_\mu$ in a weighted average over the samples $\mathcal{D}$ [Hanna et al., 2019]:

$$\hat{V}_{\text{IS}}(\pi; \hat{\mu}) := \frac{1}{m} \sum_{i=1}^m w_i r^i, \tag{1}$$

with

$$w_i := \prod_{t=0}^{t_i} \frac{p_\pi(A_t = a_t^i \mid H_t = h_t^i)}{\hat{p}_\mu(A_t = a_t^i \mid H_t = h_t^i)}. \tag{2}$$

Sufficient conditions for the estimator $\hat{V}_{\text{IS}}(\pi; \hat{\mu})$ to be an unbiased estimator of $V(\pi)$ include (sequential) *ignorability* and *overlap* [Rosenbaum and Rubin, 1983, Robins, 1986].

[1] Importance sampling estimators are often also referred to as "importance weighting" estimators.

In our setting, ignorability may be defined as for all $t$, the conditional distribution of $R$ given $A_t$ and $H_t$ is the same under $\pi$ and $\mu$, i.e., $\forall t : p_\pi(R \mid H_t, A_t) = p_\mu(R \mid H_t, A_t)$. Overlap is satisfied for a pair $(h, a)$ if it being observable under $\pi$ implies that it is observable under $\mu$, $p_\pi(A_t = a \mid H_t = h) > 0 \Rightarrow p_\mu(A_t = a \mid H_t = h) > 0$. We say that overlap is partially violated if this condition is violated for some pairs of histories and actions.

Even when ignorability and overlap are satisfied, if $\mu$ and $\pi$ differ significantly, the estimator $\hat{V}_{\text{IS}}(\pi; \hat{\mu})$ suffers from high variance. The weighted importance sampling (WIS) estimator [Rubinstein and Kroese, 2016], $\hat{V}_{\text{WIS}}(\pi; \hat{\mu}) := \frac{1}{\sum_{i=1}^m w_i} \sum_{i=1}^m w_i r^i$, introduces bias, but often has less variance. Under the Markov assumption, i.e., that context (or "state") transitions, actions and rewards depend only on the most recent context-action pair, the history $H_t$ in (2) can be replaced by $X_t$. We leave out the subscript $t$ where clear.

### 2.1 CAN WE TRUST AN IS ESTIMATE?

A fundamental challenge with off-policy evaluation is that no ground truth value, or even samples of it, is available. What is worse, the assumption of ignorability cannot be verified statistically [Rosenbaum et al., 2010] and the extent of overlap is unknown if $\mu$ is unknown. As a result, assessing the quality of an estimate $\hat{V}_{\text{IS}}$ inherently relies on domain expertise.

By examining importance weights $\{w_i\}_{i=1}^m$ and estimated propensities $\hat{p}_\mu(A_t \mid H_t)$, analysts can spot outliers with extremely large weights, and compute the effective sample size (ESS) [Gottesman et al., 2018, Owen, 2013, Chapter 9]. These practices give a per-sample and an average view of potential issues with variance and the potential for removing samples with excessive weights [Crump et al., 2009, Stürmer et al., 2010]. However, several questions remain regarding what replacing $\mu$ with $\pi$ would imply in practice:

- **Where do $\pi$ and $\mu$ differ?** How can we *describe* the inputs for which the most probable actions under $\pi$ differ from those under $\mu$?

- **If $\hat{V}(\pi) > \hat{V}(\mu)$, what gives $\pi$ the edge?** In *which* situations does acting according to $\pi$ result in higher rewards than acting according to $\mu$?

Inspecting weights and propensities in aggregate or on a per-sample basis is insufficient to answer these questions as they concern *patterns* in policy decisions, weights and rewards. For example, extreme weights and small ESS may indicate lack of overlap between $\pi$ and $\mu$ but do not explain the cause of the problem. Next, we show that a case-based model of the behavior policy $\mu$ can help identify said patterns by inducing a soft clustering over the space of histories.

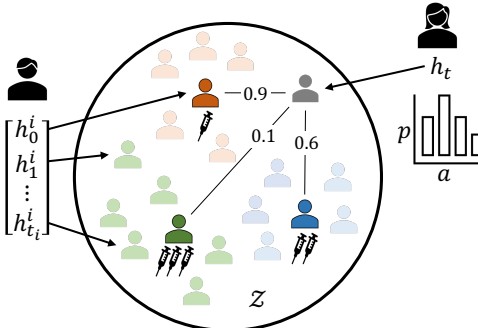

Figure 1: A schematic drawing of the prototype setup using a medical example. Each subsequence $h_t^i$ of the patient histories in the training data have a representation in the learned latent space $\mathcal{Z}$. A few subsequences are selected as prototypes—samples that are representative of the history-action space. In this example, there are three prototypes which are treated with different drug doses. Note that a patient can belong to different prototype clusters during the course of medication, as indicated with the arrows pointing out from the column vector. The action propensity $p_\mu(a \mid h_t)$ of a test sample $h_t$ is computed by weighting the similarity between $h_t$ and each prototype.

# 3 OFF-POLICY EVALUATION WITH PROTOTYPES

We propose performing off-policy evaluation using *prototype learning* [Li et al., 2018b, Ming et al., 2019]. The idea is to express the behavior policy $p_\mu(A \mid H)$ by comparing the history $H$ to a relatively small set of prototype histories from the training data, see Figure 1. In a clinical setting, such a policy may correspond to physicians choosing treatment for a new patient based on their prior experience in treating similar patients. For a domain expert, trained in interpreting such cases, a prototype-based estimate is transparent as long as the number of prototypes is small enough. By examining how policy overlap and value estimates vary with prototypes, we can answer the questions raised above.

## 3.1 MODELING BEHAVIOR WITH PROTOTYPES

Let $\tilde{H} = [\tilde{h}^1, \dots, \tilde{h}^n]^\intercal$ be a list of $n$ prototype histories.[2] Each prototype is a *subsequence* of an observed history, $\tilde{h}^j = h_t^i$ for $h^i \in \mathcal{D}$ and $t \leq t_i$. We allow the prototypes to be subsequences of full-length histories since OPE requires evaluating the behavior policy at each time step. The behavior policy $p_\mu(A_t \mid H_t = h_t)$ is approximated based on the similarity between an observation $h_t$ and the prototypes in a learned representation. The prototypes $\tilde{H}$ are themselves selected by the learning algorithm.

To learn $\tilde{H}$, we follow Li et al. [2018b], Ming et al. [2019]

[2]From now, we refer to these as "prototypes".

by first learning a set of latent prototypes as free parameters $\tilde{Z} = [\tilde{z}_1, \dots, \tilde{z}_n]^\intercal$ in an encoding space $\mathcal{Z}$. Given an encoder $e : \mathcal{H} \to \mathcal{Z}$, for an arbitrary history $h_t$, let

$$S(\tilde{Z}, e(h_t)) = [s(\tilde{z}_1, e(h_t)), \dots, s(\tilde{z}_n, e(h_t))]^\intercal$$

be the *similarity vector* for the encoding of $h_t$ comparing $e(h_t)$ to $\tilde{Z}$ using a fixed function $s : \mathcal{Z} \times \mathcal{Z} \to \mathbb{R}_+$. We use an RBF-kernel with unit bandwidth ($\gamma = 1$),

$$s(\tilde{z}, e(h)) \coloneqq \exp(-\|\tilde{z} - e(h)\|_2^2 / \gamma^2), \quad (3)$$

which takes values between 0 (no similarity) and 1 (full similarity). With $B \in \mathbb{R}^{k \times n}$, we estimate the behavior policy $\mu$ through logistic regression in the space induced by $S$,

$$\hat{p}_\mu(A_t \mid H_t = h_t) = f_\sigma(BS(\tilde{Z}, e(h_t)) + c), \quad (4)$$

where $f_\sigma$ denotes the softmax function over rows and $c \in \mathbb{R}^k$ is a bias term. Column $j$ of $B$ represents the coefficients determining the action probabilities associated with $\tilde{h}^j$. If the coefficient $B_{ij}$ is positive, higher similarity between $h_t$ and $\tilde{h}^j$ makes action $i$ more probable for $h_t$; a negative coefficient makes action $i$ less probable.

The model parameters $\Theta = (e, B, c, \tilde{H})$, comprising the parameters of the encoder $e$, coefficients $B$, $c$ and the set of prototypes $\tilde{H}$, are all unknown and must be learned from data. As encoder, we use either feedforward or recurrent neural networks. Following Ming et al. [2019], we learn $\Theta$ by minimizing the regularized negative log-likelihood (NLL)

$$J(\Theta) = \mathrm{NLL}(\mathcal{D}; \Theta) + \lambda_d R_d(\Theta) + \lambda_c R_c(\Theta) + \lambda_e R_e(\Theta) \quad (5)$$

using stochastic gradient descent. The regularization terms $R_d(\Theta)$, $R_c(\Theta)$ and $R_e(\Theta)$ encourage *diversity*, *clustering* and *evidence*, respectively, and are defined in Appendix A.

To make sure that prototypes represent real cases, i.e., to select $\tilde{H}$, latent prototypes are projected onto encodings of training samples at regular intervals between descent steps,

$$\tilde{h}^j \leftarrow \underset{h_t^i \in \overline{\mathcal{D}}}{\arg\max} \, s(\tilde{z}_j, e(h_t^i)) \quad \text{and} \quad \tilde{z}_j \leftarrow e(\tilde{h}^j), \quad (6)$$

with $\overline{\mathcal{D}}$ the set of all subsequences of trajectories in $\mathcal{D}$.

**Is there a good prototype model?** Modeling the behavior policy using prototypes places additional restrictions on the functional form of estimates. It is natural to ask: Assuming that adjusting for the history $H_t$ is sufficient for unbiased policy evaluation, do there exist prototype histories $\tilde{H}$, an encoding $e$ and a similarity function $s$ such that evaluation using the prototype model is exact or accurate? In Section 4, we study this question empirically. Additionally, in Appendix A.2, we show constructively that there are indeed problems for which a prototype model exists that *exactly* describes the behavior policy $\mu$.

## 3.2 PREDICTING WITH PROTOTYPES

When computing the estimated behavior policy (4) for a history $h$, the similarity vector $S(e(\tilde{H}), e(h))$ determines how similar each of the $n$ prototypes are to $h$. The number $n$ is a hyperparameter. The more prototypes are used, the greater the flexibility of the model, but a large $n$ may result in $S$ consisting of multiple elements close to 1, making predictions difficult to interpret. For example, if $s(e(\tilde{h}^j), e(h)) \approx 1$ for more than 10 prototypes $j$, it may be difficult to reason about the policy decision after all.

To address this, we use only a limited number of $q \leq n$ prototypes—so-called *prediction prototypes*—when making predictions with the *trained* model. Let $s_q(h)$ be the similarity between $e(h)$ and its $q$th most similar latent prototype. For $j = 1, \ldots, n$, we truncate the similarity vector according to

$$s(\tilde{z}_j, e(h)) \leftarrow \begin{cases} s(\tilde{z}_j, e(h)) & \text{if } s(\tilde{z}_j, e(h)) \geq s_q(h), \\ 0 & \text{otherwise.} \end{cases}$$

As an example, with $q = 2$ the (sorted) similarity vector in Figure 1 would become $[0.9, 0.6, 0]^\intercal$.

We perform the truncation step independently for all contexts $h$. In Section 4.2, we study the resulting trade-off between transparency (small $q$ and $n$) and bias as we vary the number of (prediction) prototypes, $(q)$ $n$. Note that we optimize the regularization parameters $\lambda_d$, $\lambda_c$ and $\lambda_e$ with respect to the choice of $q$.

## 3.3 USING PROTOTYPES FOR EVALUATION

Prototypes induce a soft clustering of the space of histories. Each prototype represents a group of similar histories which can be associated with a certain distribution over actions. In Figure 1, we see for example that the "green prototype"— representing the patients in the "green cluster"—is given a higher dose of the drug than the other prototypes. Given characteristics of the "green prototype", a domain expert should be able to explain why it receives this type of treatment. While it is possible to use other methods to cluster the space of histories, prototypes have the advantage of being based in cases and trained to describe groups of subjects who are treated differently under the behavior policy. We see in Section 4.2 that this is beneficial also for accuracy.

When modeling the behavior policy $\mu$ using prototypes, we can utilize the induced clustering structure to answer the questions raised in Section 2.1. First of all, the prototypes $\tilde{h}^j$ and their action probabilities

$$\hat{p}_\mu^j(a) = \hat{p}_\mu(A = a \mid H = \tilde{h}^j) \qquad (7)$$

give an overview of the estimated behavior policy. By comparing the action probabilities $\hat{p}_\mu^j(a)$ with the corresponding action probabilities under $\pi$, $p_\pi^j(a) = p_\pi(A = a \mid H = \tilde{h}^j)$,

we can explain input regions for which $\pi$ and $\mu$ differ in their suggested actions. Domain experts can use this overview to assess how well the data supports evaluation of the target policy. For example, if $\pi$, for a certain prototype, suggests actions that are extremely rare under $\mu$, there may not be enough data on these decisions to accurately estimate $V(\pi)$.

It is good practice to compare $\hat{V}(\pi)$ with $\hat{V}(\mu)$, i.e., the mean reward in data. If $\hat{V}(\pi)$ is different from $\hat{V}(\mu)$, we would like to know for which inputs $\pi$ gain or lose performance in relation to $\mu$. The prototypes allow us to divide the estimated values into prototype-based contributions and answer this question. To make the implicit clustering explicit, we define $J_t$ to be a random variable with values in $\{1, \ldots, n\}$, representing an assignment of a history $H_t$ to prototype $j$ at time $t$. We let the probability of being assigned to prototype $j$ be proportional to the similarity $s$,

$$p(J_t = j \mid H_t = h_t) = \frac{s(\tilde{z}_j, e(h_t))}{\sum_{k=1}^n s(\tilde{z}_k, e(h_t))}. \qquad (8)$$

Now, we define the value $V_{j,t}(\pi)$ of prototype $j$ at time $t$, obtained under a policy $\pi$, as the expected future reward under $\pi$ given the assignment $J_t = j$:

$$V_{j,t}(\pi) := \mathbb{E}_\pi[R_T \mid J_t = j]. \qquad (9)$$

With $p(J_t = j)$ the marginal probability of being assigned to prototype $j$ at time $t$, by the law of total expectation, $V(\pi) = \sum_{j=1}^n V_{j,t}(\pi)p(J_t = j)$ for any $t$. Each term

$$V_{j,t}(\pi)p(J_t = j) \qquad (10)$$

in the sum represents the contribution to the overall value $V(\pi)$ from histories which are similar to prototype $j$ at time $t$, effectively stratifying the value by types of situations. Note that we can compute $V_{j,t}(\mu)$ in a similar way to compare the estimated values of $\pi$ and $\mu$ from a prototype perspective.

We may express $V_{j,t}(\pi)$ as a weighted expectation under the behavior policy $\mu$, with importance weights $W$,

$$V_{j,t}(\pi) := \mathbb{E}_\mu \left[ \frac{p(J_t = j \mid H_t = h_t)}{p_\pi(J_t = j)} W R_T \right],$$

where $p_\pi(J_t)$ is found by importance-weighted marginalization over $H_t$ (see derivation in Appendix A.1). We use this strategy to estimate $V_{j,t}(\pi)$ from finite samples.

## 4 EXPERIMENTS

In Section 4.1, we illustrate our method by examining an example of sepsis management. Using patient data from the MIMIC-III database [Johnson et al., 2016], we evaluate a replication of the so-called AI Clinician implemented by Komorowski et al. [2018], see below for details. In Section

4.2, we inspect the prototype model in more detail. We study the trade-off between transparency and bias and compare the model to several baseline estimators. By utilizing a sepsis simulator, we also investigate the bias induced by prototypes as a function of the sequence length.

**AI Clinician.** The AI Clinician is a clinical decision support model for sepsis management [Komorowski et al., 2018]. The model is learned from data of sepsis patients extracted from the MIMIC-III database [Johnson et al., 2016]. The patient data (e.g., demographics, vital signs and laboratory values) are coded as multidimensional time series with a discrete time step of 4 hours. There are two treatment variables: the total volume of intravenous (IV) fluids ($f$) and maximum dose of vasopressors ($v$) administered over each 4-hour period. In short, the AI Clinician is built by clustering the data into 750 states, discretizing the combinations of treatment doses into 25 possible actions $(f, v) \in \{0, 1, 2, 3, 4\}^2$, and solving the corresponding Markov decision process using value iteration. The final reward $r^i$ is $+100$ if the patient survived and $-100$ if the patient died. This process is repeated 500 times, each time with a new train-test split, and the model with the highest WIS value estimate on the test set is taken as the target policy, $\pi_{\text{AIC}}$. We use the data split associated with $\pi_{\text{AIC}}$ in our experiments.

**Experimental setup.** We consider two types of encoders for the prototype framework: a feedforward neural network (FNN) and a recurrent neural network (RNN). Both encoders have two layers of size 64 with ReLu and tanh, respectively, as activation function. We name these models ProNet and ProSeNet, respectively. We compare the prototype framework to several baseline models: a logistic regression classifier (LR), a random forest classifier (RF), a vanilla FNN, a vanilla RNN, and a model based on post-hoc clustering of RNN-encoded histories. The neural network baselines have the same structure as the corresponding prototype encoder. In the main sepsis experiment, we train all neural networks over 400 epochs, using a batch size of 64 for RNN and ProSeNet, and 1024 for FNN and ProNet. We use the Adam algorithm for optimization with learning rate 0.001, weight decay 0.001 and otherwise default parameters. Furthermore, we use the NLL loss when training the vanilla neural networks. All models are calibrated using sigmoid calibration on a held-out validation set (25 % of the training data). Further details, including hyperparameter selection, are provided in Appendix B.[3]

## 4.1 DEMONSTRATING THE FRAMEWORK

To demonstrate the benefit of learning prototypes in OPE, we estimate the behavior policy $\mu$, i.e., the policy followed

---

[3]The code is available at `https://github.com/Healthy-AI/case_based_ope`.

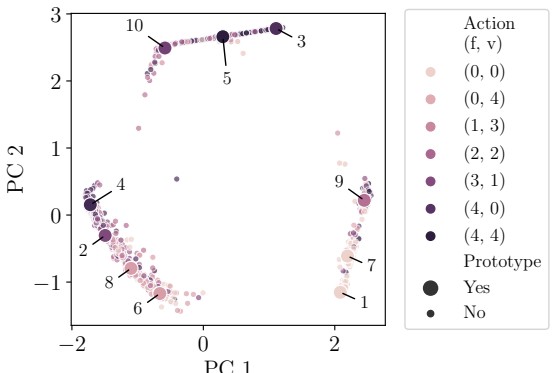

Figure 2: A PCA plot of encoded training data, colored w.r.t. the action (dose of IV fluids and vasopressors) taken by the physicians. The prototypes are numbered 1–10. Note that prototype learning affects the structure of the encoding space. Post-hoc clustering of the latent space of a model trained without prototypes gives a substantially worse approximation of the behavior policy, see Table 1.

by clinicians in the MIMIC-III data, using a prototype model with $n = 10$ prototypes, $q = 2$ prediction prototypes and an RNN encoder (i.e., a ProSeNet model). As an overview of the relationship between prototypes, a PCA plot of encoded training data is shown in Figure 2. The latent prototypes are numbered 1–10 and the colors indicate action chosen by $\mu$ in the data. Note that the figure is intended to help the reader to orient him/herself in this section; we do not rely on this projection in itself.

We can interpret the prototypes by visualizing trajectories of the corresponding patients. In Figure 3, we take a closer look at prototypes 5, 7 and 8, which represent each of the major clusters in Figure 2. By plotting three key features—heart rate (HR), mean blood pressure (BP) and SOFA score—and the treatment variables against time, we immediately get a sense of which type of patients the prototypes represent. For example, the patient corresponding to prototype 5 has high heart rate, low blood pressure and high SOFA score—signs of severe sepsis—and receives an aggressive treatment.[4] The prototype 7 patient, who has lower heart rate, higher blood pressure and lower SOFA score, receives low doses of IV fluids and vasopressors.

We understand that trajectories that are most similar to prototype 7 in the latent space belong to patients who are currently relatively healthy, which they are more likely to be at an early stage of the course. Interestingly, we observe that all encoded histories until time $t = 0$ are most similar to either prototype 7 or prototype 9. Is it therefore relevant to ask the question: If we were to follow the AI Clinician instead of the behavior policy, how would the treatment strategy

---

[4]SOFA is an abbreviation for sequential organ failure assessment.

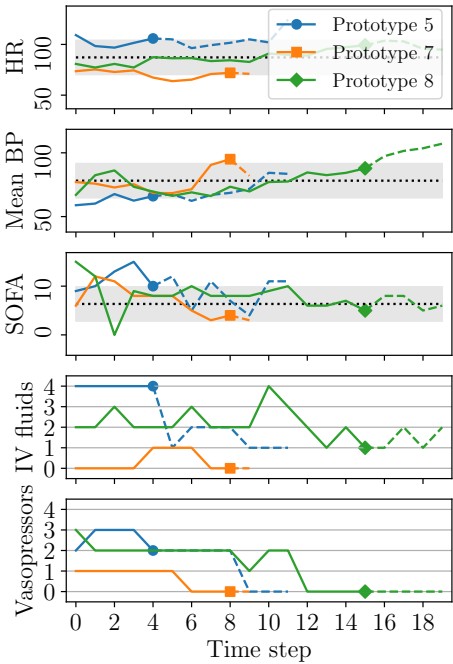

Figure 3: Vital signs and SOFA score plotted against time for three different prototype patients (upper three panels). The dashed black lines show the data average of each feature and the shaded areas mark $\pm 3$ standard deviations. The lower two panels show the actions taken by physicians. The time index of each prototype subsequence is marked with filled marker; for example, prototype 5 is the subsequence ending at time step 4 of the corresponding patient history.

change in an initial stage? That is, how do $\mu$ and $\pi_{AIC}$ differ for prototypes 7 and 9?

We can answer this question by comparing the distributions of actions taken under $\mu$ and $\pi_{AIC}$ for these prototypes, see Figure 4. For prototype 7, we see that the most likely treatment under both $\mu$ and $\pi_{AIC}$ is to not give any IV fluids or vasopressors. However, under $\pi_{AIC}$, there is also a relatively high probability of increasing the dose of IV fluids—a rare action under $\mu$. The differences are even greater for prototype 9, where $\pi_{AIC}$ has a nonzero probability of giving an aggressive treatment with combinations of IV fluids and vasopressors. Under $\mu$, these treatments have almost zero probability. A domain expert can reason about the validity of $\pi_{AIC}$: given characteristics of the patient corresponding to prototype 9, would it be medically sound to treat this type of patient as suggested by $\pi_{AIC}$ in Figure 4?

From an OPE perspective, the initial differences between $\mu$ and $\pi_{AIC}$ make it difficult to accurately estimate $V(\pi_{AIC})$. A known problem with importance sampling is that the variance of the weights $w_i$ can grow exponentially with the sequence length [Liu et al., 2018]. Here, the assumption of overlap is potentially partially violated already at the first time step, and regardless of model of the behavior policy,

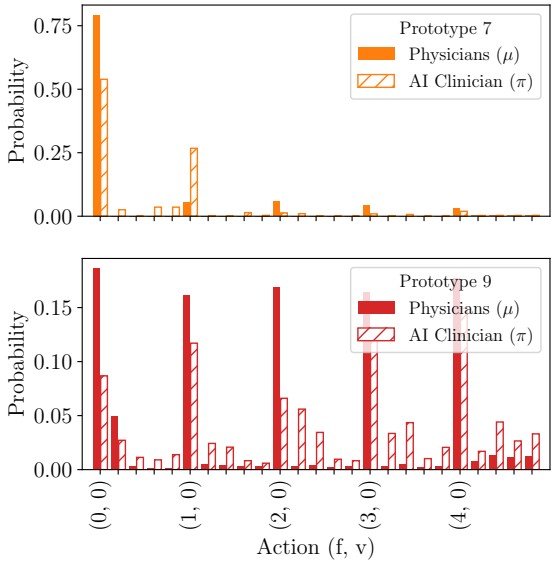

Figure 4: Action distribution under $\mu$ and $\pi_{AIC}$ for prototype 7 and 9. For $\mu$ the probabilities are computed according to (7). Since $\pi_{AIC}$ is deterministic, we normalize the distribution of actions suggested by $\pi_{AIC}$ for input histories that are most similar to respective prototype in the latent space.

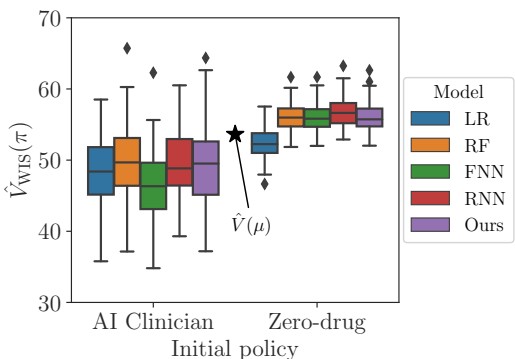

Figure 5: Bootstrap estimates of the value of the target policy of following the AI Clinician and the zero-drug policy, respectively, for one time step and then following the behavior policy. The estimated value of the behavior policy $\mu$, $\hat{V}(\mu)$, is included as a reference.

we observe high variance in the IS weights (ranging from $\ll 1$ to the order of $10^3$) and an extremely small effective sample size ($< 10$). To reduce variance, we instead evaluate a different target policy where we follow $\pi_{AIC}$ in the first time step and then follow $\mu$ until the end of the sequence. For comparison, we do the same with a zero-drug policy $\pi_0$ which suggests leaving patients untreated.

Figure 5 shows 100 bootstrap estimates of the policy values using five different estimators of $\mu$: the baselines LR, RF, FNN and RNN, and our prototype model. For all models except the RNN baseline and the prototype model, we make

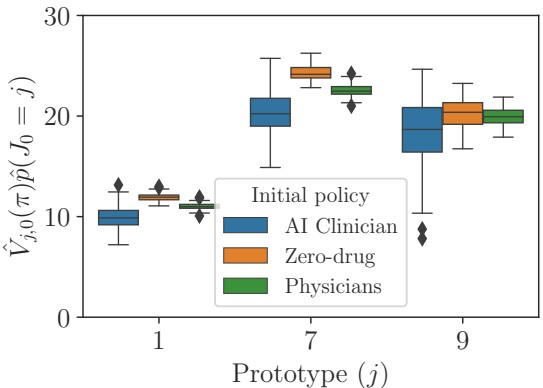

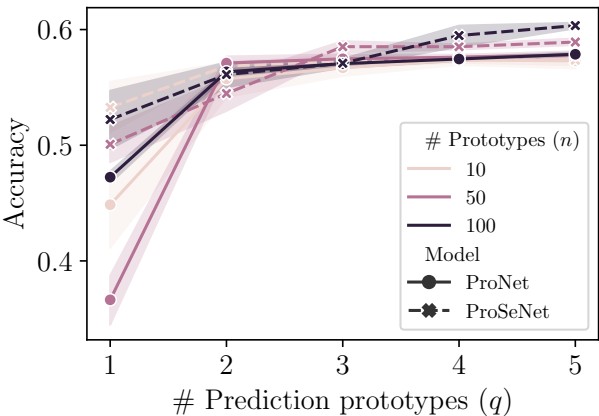

Figure 6: Bootstrap estimates of $V_{j,0}(\pi)p(J_0 = j)$ for prototype 1, 7 and 9. As described in Section 3.3, $V_{j,t}(\pi)p(J_t = j)$ is the the value of prototype $j$ at time $t$ multiplied with the marginal probability of being assigned to prototype $j$ at time $t$.

Figure 7: Accuracy on the sepsis test data for ProNet and ProSeNet using a varying number of (prediction) prototypes ($q$) $n$. The setting with $n = 10$ and $q = 2$ works well here.

the Markov assumption and model $p(A \mid H)$ using only the last context-action pair of the history. We note that the results are consistent across estimators. In comparison with the estimated value of $\mu$, $\hat{V}(\mu)$, the results indicate that it could be beneficial to avoid giving drugs to patients at the initial time step, while it seems less favorable to follow $\pi_{\text{AIC}}$.

The prototypes allow us to break down the result and answer the question: If $\hat{V}(\pi) \neq \hat{V}(\mu)$, where does $\pi$ gain or lose performance? In Figure 6, we show 100 bootstrap estimates of the contribution to the overall value, see (10), for prototypes 1, 7 and 9 at time $t = 0$. These prototypes define the bottom right cluster of Figure 2 where all encoded histories until time $t = 0$ belong. As expected, prototype 7 and 9 contribute the most to the overall value. Note that the difference in variance between these prototypes is explained by Figure 4 where $\pi_{\text{AIC}}$ and $\mu$ differ more for prototype 9 than for prototype 7. Also note that the trend in Figure 5 is repeated here: following the zero-drug policy at the initial time step is generally better than following the AI Clinician.

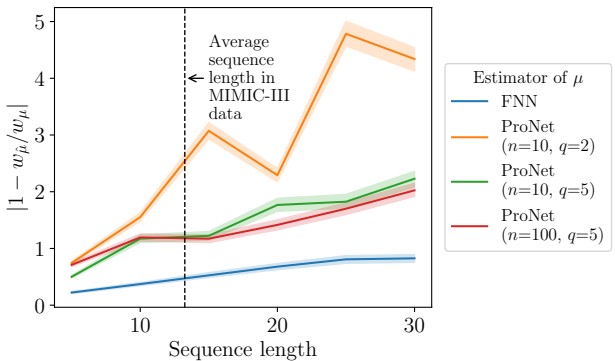

Figure 8: The relative error of estimated importance weights for increasing sequence lengths in the sepsis simulator. Ideally, if $\hat{\mu}$ is a good estimator of $\mu$, the ratio $w_{\hat{\mu}}/w_{\mu}$ should be equal to 1. We see that approximating $\mu$ with prototype models gives rise to a bias in relation to modeling $\mu$ with a plain FNN.

100 is small. Using only two prediction prototypes works well for this dataset.

In Table 1, we compare the prototype models with $n = 10$ and $q = 2$ to the baseline estimators in approximating $p_{\mu}(A \mid H_t)$. Here, we report accuracy, the area under the ROC curve (AUC) and the static calibration error (SCE) [Nixon et al., 2019], a multiclass extension of the expected calibration error. The prototype models are superior to the (regularized) LR but they perform slightly worse than the black-box models RF, FNN and RNN. However, as we see in Figure 7, with increased number of prototypes, ProSeNet has the capacity to approach the performance of these models, at least in terms of accuracy. Finally, we note that the prototype models are superior to a model where post-hoc clustering of the RNN encodings are used to identify "prototypes", showing the power of learning prototypes

## 4.2 PERFORMANCE OF THE PROTOTYPE MODEL

While increasing transparency, the use of prototypes imposes restrictions on the model, possibly increasing the approximation error. In Figure 7, we show the accuracy of two different prototype models—ProNet and ProSeNet—in approximating $p_{\mu}(A \mid H_t)$ on the sepsis test data for a varying number of prototypes $n$ and prediction prototypes $q$ (see Section 3.2). As encoder, the models use an FNN (ProNet) and an RNN (ProSeNet), respectively. The sequential model, making use of the entire history $H_t$, performs the best, especially for $q = 1$ and $q \geq 4$. Interestingly, the effect of increasing the number of prototypes from 10 to 50 or even

in a supervised manner.

In practice, the trade-off between transparency and bias is likely less of a problem. In the process of diagnosing policy value estimates, we may sacrifice some accuracy in favor of interpretability. That is, we can learn a model with few prototypes to reason about the target and behavior policies. Then, if the initial analysis indicates that the data supports evaluation of the target policy, we can learn a more complex model for the actual policy evaluation.

#### 4.2.1 Bias Due to Increased Sequence Length

If the use of prototypes introduces a bias in the estimated propensity, it is natural to ask what it means for the sequential setting, where multiple propensities are multiplied together to form the importance weights. To quantity this effect, we consider the synthetic environment of sepsis management provided by Oberst and Sontag [2019]. By sampling a large amount of data from the environment, we estimate the true parameters of the underlying Markov decision process. We then learn an optimal behavior policy using policy iteration. We refer to Appendix B.2 for details.

We collect trajectories of the behavior policy of various lengths, from 5 to 30 time steps. For each trajectory length, we use the data to (a) estimate the behavior policy $\mu$ using a vanilla FNN and FNN-based prototype models with varying number of (prediction) prototypes, respectively; (b) learn a target policy $\pi$ using policy iteration; and (c) estimate the value of $\pi$ using both the true behavior policy $\mu$ and its estimators $\hat{\mu}$. Note that any difference in the estimated values stems from a difference in the importance weights $w$. In Figure 8, we plot the relative error of estimated weights against the trajectory length for four estimators of $\mu$.[5] Averaging over 100 iterations of the sampling and learning process, we see that the ratio $w_{\hat{\mu}}/w_{\mu}$ generally differs from 1 for all estimators and that modeling with prototypes induces larger bias than using an FNN. For longer sequences, the number of prediction prototypes $q$ becomes critical.

Finally, we quantify the absolute effect prototypes have on the value estimate for sequences of length 15, which is close to the average sequence length in the MIMIC-III data. We compute the true value of $\pi$ by running it in the simulator and compare this value to weighted IS estimates using the estimators in Figure 8. We observe final rewards $r^i = \pm 1$ if a simulated patient is discharged or dies; otherwise $r^i = 0$. Averaging over 100 iterations, the estimated value has an absolute difference from the true value that amounts to 0.39 for the FNN (standard deviation 0.27), 0.46 (0.33) for ProNet with $n = 10$ and $q = 2$, 0.50 (0.34) for ProNet with $n = 10$ and $q = 5$ and 0.41 (0.31) for ProNet with $n = 100$ and $q = 5$. These results should be compared to

---

[5]Note that the probabilities under $\pi$ cancel when considering the ratio of the weights.

the average value of $\pi$ (0.06 (0.10)) and the WIS estimate using the true behavior policy (0.38 (0.27)).

## 5 RELATED WORK

Issues with importance sampling methods for OPE are well known. Several works aim at describing issues related to high variance [Gottesman et al., 2019], or mitigating them using methodological advances [Precup, 2000, Thomas and Brunskill, 2016, Jiang and Li, 2016, Schneeweiss et al., 2009, Swaminathan and Joachims, 2015]. Others aim to use weights to identify a new study population for which the policy's value can be more efficiently estimated [Li et al., 2018a, Fogarty et al., 2016]. Oberst et al. [2020] emphasize the value of interpretability in this endeavour to communicate the generalizability of the estimate. Our method is compatible with all three approaches, allowing for transparent descriptions of variance issues, identifying new study populations and for use as plug-in estimates.

Interpretability is a an important component of learning systems deployed in increasingly critical functions [Rudin, 2019, Lipton, 2018]. Rule-based estimators, such as rule list [Wang and Rudin, 2015] and decision trees, are often favored for their short descriptions but generalize poorly to sequential inputs which are the focus of this work. Gottesman et al. [2020] proposed an approach for interpretable OPE which highlights transitions in data whose removal would have a large impact on the estimate. This approach is related to ours but answers a different set of questions.

Evaluating policies using direct sample-to-sample comparison has a long tradition in policy evaluation through the use of matching estimators of causal effects, see e.g., [Rosenbaum and Rubin, 1983, Rubin, 2006, Kallus, 2020]. While favored for its transparency, this approach is typically only used to compare two deterministic policies such as "treat all" or "treat none". Matching often relies either on specifying a similarity function in advance or on an estimate of the behavior policy. In high-dimensional settings, this often leads to bias or lost interpretability. Our approach aims to combine the transparency of matching estimators with the flexibility of representation learning methods.

## 6 CONCLUSION

In this work, we have studied off-policy evaluation (OPE) using importance sampling (IS) in the case where the behavior policy $\mu$ is unknown and must be estimated from data. While IS is a popular OPE method, it may be difficult to assess the quality of an IS value estimate. Standard practices, such as inspecting importance weights, provide only an average or a per-sample view of potential issues. To address this issue, we proposed estimating the behavior policy for IS using prototype learning to better explain patterns in policy

Table 1: A summary of performance on the sepsis test data for different estimators of the behavior policy $p_\mu(A \mid H_t)$. For ProNet and ProSeNet, $n = 10$ and $q = 2$. The 95 percent confidence intervals are calculated from 1000 bootstraps.

| Model | Accuracy ($\uparrow$) | SCE ($\downarrow$) | AUC ($\uparrow$) |
|---|---|---|---|
| LR | 0.38 (0.38, 0.39) | 0.0112 (0.0110, 0.0115) | 0.88 (0.88, 0.88) |
| RF | 0.62 (0.61, 0.62) | 0.0037 (0.0034, 0.0039) | 0.93 (0.93, 0.93) |
| Post-hoc clustering | 0.44 (0.44, 0.45) | 0.0097 (0.0096, 0.0101) | 0.86 (0.85, 0.86) |
| FNN | 0.61 (0.61, 0.61) | 0.0041 (0.0039, 0.0044) | 0.93 (0.92, 0.93) |
| ProNet ($n = 10$, $q = 2$) | 0.56 (0.55, 0.56) | 0.0069 (0.0067, 0.0072) | 0.90 (0.90, 0.90) |
| RNN | 0.62 (0.62, 0.63) | 0.0056 (0.0053, 0.0058) | 0.94 (0.94, 0.94) |
| ProSeNet ($n = 10$, $q = 2$) | 0.57 (0.57, 0.58) | 0.0057 (0.0054, 0.0059) | 0.91 (0.91, 0.91) |

decisions and value estimates. We demonstrated our idea using a real-world example of sepsis management. While the use of prototypes increases the approximation error, we found that prototype models have the capacity to perform similarly to plain neural networks.

When reflecting upon the results of the sepsis study it may seem strange that it would be advantageous to never treat patients at the onset of sepsis. Even though prototypes serve as a tool for inspecting policies and value estimates, they do not answer all questions. For example, there may be variables affecting both the treatment and the outcome that are not present in the observed data. In such a case, the ignorability assumption defined in Section 2 no longer holds. Failure to detect such an issue is not a limitation of prototypes; ignorability cannot be verified by statistical means.

A limitation of our analysis is that it does not separate different types of errors introduced by prototype learning. We conjecture that a model with fewer prototypes is more likely to overestimate overlap between behavior and target policies, rather than underestimate it, due to increased smoothness in the estimated behavior policy. This will likely result in less extreme importance weights and lower variance, potentially at the cost of increased bias. We hope to provide analysis which more precisely characterizes the approximation error as a function of the number of prototypes in future work.

Finally, we have used subsequences of all variables in patient histories as prototypes. This choice is aligned with the literature on sequence prototypes but is not the only option. For example, in applications of prototype learning to image classification, parts of images were used as prototypes [Li et al., 2018b], not entire images from the training set. To simplify description further and improve interpretability in policy evaluation, we may define prototypes as sequences of only variables which are important for the behavior policy.

## Acknowledgements

We would like to thank Patrick Royer for insightful discussions regarding the sepsis management policy evaluation. Furthermore, we would like to thank Devdatt Dubhashi, Emil Carlsson, Morteza Haghir Chehreghani and Adam Breitholtz for valuable feedback on this work.

This work was partially supported by the Wallenberg AI, Autonomous Systems and Software Program (WASP) funded by the Knut and Alice Wallenberg Foundation.

The computations in this work were enabled by resources provided by the Swedish National Infrastructure for Computing (SNIC) at Chalmers Centre for Computational Science and Engineering (C3SE) partially funded by the Swedish Research Council through grant agreement no. 2018-05973.

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
