# OpenReview forum: "Case-Based Off-Policy Evaluation Using Prototype Learning"
_auai.org/UAI/2022/Conference — UAI 2022 Poster_

### Official Review · Reviewer_ix6i · 2022-03-31

**Q2(1) Originality/Novelty:** 3
**Q2(2) Significance/Impact:** 3
**Q2(3) Correctness/Technical Quality:** 3
**Q2(6) Clarity Of Writing:** 4
**Q6 Overall Score:** 6
**Q8 Confidence In Your Score:** 4

**Q1 Summary And Contributions:**

This paper use prototype learning to model the behavior policy and estimate the policy value of the target policy. The employment of prototype learning improve the interpretability of the method and answer the two question: 1. Where do $\pi$ and $\mu$ differ? 2. What give $\pi$ the edge. The experimental results on sepsis management gives some inspiring insight on the analysis of target policy.

**Q2 Assessment Of The Paper:**

More detailed information regarding each of these aspects is given below:

**Q2(4) Quality Of Experiments (Optional):**

3: Good: The experimental evaluation is adequate, and the results convincingly support the main claims.

**Q2(5) Reproducibility:**

3: Good: Key resources (e.g., proofs, code, data) are available and key details (e.g., proofs, experimental setup) are sufficiently well-described for competent researchers to confidently reproduce the main results.

**Q3 Main Strengths:**

1. The paper incorporates the prototype learning to offline policy evaluation, which brings impressive interpretability.
2. The experimental results is sufficient. The author also conduct experiments to analyze the influence of the number of prototypes, prediction-prototypes on the results.
3. The paper is well written and convenient to follow. The information about the algorithm is described in a detailed way.

**Q4 Main Weakness:**

1. The method in the paper somehow lacks novelty. It seems to be a direct combination of offline policy evaluation and prototype learning.
2. The performance of the method is not competive enough. The baselines performs better than it.
3. The section of experiments does not describe all the necessary information of the experimental settup. I think the authors could improve it and give more detailed information.

**Q5 Detailed Comments To The Authors:**

1. I think the policy value is a continuous value rather than a binary variable. Therefore, I am confused about how to compute the accuracy and AUC of the method.

**Q7 Justification For Your Score:**

I carefully read the paper for over 5 hours and obtain a comprehensive understand of the paper.

**Q9 Complying With Reviewing Instructions:**

1: Yes.

---

### Official Review · Reviewer_n4At · 2022-04-12

**Q2(1) Originality/Novelty:** 3
**Q2(2) Significance/Impact:** 3
**Q2(3) Correctness/Technical Quality:** 3
**Q2(6) Clarity Of Writing:** 4
**Q6 Overall Score:** 7
**Q8 Confidence In Your Score:** 2

**Q1 Summary And Contributions:**

The paper presents a method for evaluating a target policy based on a set of prototype trajectories that are learned from observations collected according to a behavioral policy. The main advantage of this approach is its interpretability although performance may be reduced compared to black box models.

**Q2 Assessment Of The Paper:**

More detailed information regarding each of these aspects is given below:

**Q2(4) Quality Of Experiments (Optional):**

3: Good: The experimental evaluation is adequate, and the results convincingly support the main claims.

**Q2(5) Reproducibility:**

2: Fair: Key resources (e.g., proofs, code, data) are unavailable but key details (e.g., proof sketches, experimental setup) are sufficiently well-described for an expert to confidently reproduce the main results.

**Q3 Main Strengths:**

The paper is well written and technically sound. The idea presented are an interesting contribution to explainable AI.
Experiments illustrate well the methods, discussing both to its strengths and weaknesses.


**Q4 Main Weakness:**

The application to the AI Clinician shows that the possibility to apply the method only if the behavioral and target policy are similar is quite a strong limitation. Also, the loss in accuracy, compared to plain neural networks is not neglectable.

**Q5 Detailed Comments To The Authors:**

At page 2, just after eq. (3), R^{k x n} --> What is k?
At page 5 the short description of the AI Clinician approach was not clear to me. Besides, you mention 25 possible actions, but in figure 2 I see only 7.


**Q7 Justification For Your Score:**

The interpretability of the method makes it very interesting, despite the reduce performance, in some fields such a the medical one.

**Q9 Complying With Reviewing Instructions:**

1: Yes.

---

### Official Review · Reviewer_AJdH · 2022-04-13

**Q2(1) Originality/Novelty:** 2
**Q2(2) Significance/Impact:** 2
**Q2(3) Correctness/Technical Quality:** 3
**Q2(6) Clarity Of Writing:** 3
**Q6 Overall Score:** 5
**Q8 Confidence In Your Score:** 4

**Q1 Summary And Contributions:**

Importance sampling(IS) has been commonly used in OPE literature due to its simplicity. However, it can be problematic when the behavior policy is unknown. The paper proposes to improve IS by prototype learning, which has been developed in previous literature. Experiments on sepsis management have been conducted. The learned prototypes improve interpretability and provide new insights. The paper also studies the bias by using prototype learning and how that affects the IS estimator.

**Q2 Assessment Of The Paper:**

More detailed information regarding each of these aspects is given below:

**Q2(4) Quality Of Experiments (Optional):**

3: Good: The experimental evaluation is adequate, and the results convincingly support the main claims.

**Q2(5) Reproducibility:**

3: Good: Key resources (e.g., proofs, code, data) are available and key details (e.g., proofs, experimental setup) are sufficiently well-described for competent researchers to confidently reproduce the main results.

**Q3 Main Strengths:**

-While most previous literature has been focused on improving the performance of off-policy evaluation estimators, interpretability has not been studied a lot in this setting, which makes the setting of this paper rather novel.

-The paper provided a thorough go-through of how to interpret the result from the prototypical network in the experiment setting, with systematic studying of different design choices of their method, how increased sequence length could introduce bias, etc.


**Q4 Main Weakness:**

The paper applies the prototypical network to the task of off-policy evaluation. The main concern I have about this paper is how to justify this combination. Prototypical networks as in [Li 2018b] are originally proposed to improve the interpretability of classification tasks in the iid setting without taking distribution shifting into consideration. While one of the most important issues about off-policy evaluation is that this is a distribution shifting problem. For example, IS method is famous for suffering from high variance, especially in the setting when behavior policy is unknown, how could using a prototypical network instead relieve the high-variance issue? Also, using prototypical networks will introduce new hyper-parameters, such as the number of prototypes and the number of prediction prototypes to use, which could also be problematic in practice.

**Q5 Detailed Comments To The Authors:**

-In the experiment part on Page 6, the paper says for all models except the RNN and prototype model, we make the Markov assumption,  could this be elaborated more? Will this create an unfair comparison?

**Q7 Justification For Your Score:**

While I agree interpretability is under-explored in the ope setting, some more explanations need to be made to justify the use of prototypical networks in the ope setting.

**Q9 Complying With Reviewing Instructions:**

1: Yes.

---

### Official Review · Reviewer_5ABX · 2022-04-13

**Q2(1) Originality/Novelty:** 3
**Q2(2) Significance/Impact:** 3
**Q2(3) Correctness/Technical Quality:** 3
**Q2(6) Clarity Of Writing:** 3
**Q6 Overall Score:** 6
**Q8 Confidence In Your Score:** 3

**Q1 Summary And Contributions:**

This paper proposes the use of prototype learning as a diagnostic tool for off-policy evaluation. In particular, the authors propose using prototypes as a way to provide human interpretable diagnostics around where two policies differ, and where the rewards of the two policies may differ. It argue IS weights are inappropriate for this task due to their lack of explainability, then describe the use of prototype learning. Experimental results show interpretability, and the relative efficacy.

**Q2 Assessment Of The Paper:**

More detailed information regarding each of these aspects is given below:

**Q2(4) Quality Of Experiments (Optional):**

3: Good: The experimental evaluation is adequate, and the results convincingly support the main claims.

**Q2(5) Reproducibility:**

3: Good: Key resources (e.g., proofs, code, data) are available and key details (e.g., proofs, experimental setup) are sufficiently well-described for competent researchers to confidently reproduce the main results.

**Q3 Main Strengths:**

1. Compelling problem--interpretability is an important aspect of off-policy evaluation when humans are likely to be in the loop.
2. Simple, practical solution which is well motivated
3. Strong empirical results.

**Q4 Main Weakness:**

1. No theoretical contribution/examination of proposed approach
2. No clear statement of necessary assumptions

**Q5 Detailed Comments To The Authors:**

Overall, I liked this paper and think it is an interesting approach to an important problem. The solution is compelling both in its simplicity and its apparent efficacy. The one thing I would like to have seen is some kind of theoretical examination of the properties of the proposed estimator. Because the task is inherently unsupervised, it is important for practitioners to be able to reason over the assumptions required for the proposed approach and also possible worst case behavior. This, arguably, is one of the strengths of IS which has been incredibly well studied for decades.

**Q7 Justification For Your Score:**

Overall, I think this paper is interesting and would be happy to see it published, but I would have liked to also have seen an examination beyond empirical results.

**Q9 Complying With Reviewing Instructions:**

1: Yes.

---

### Decision · Program_Chairs · 2022-05-15

**Decision:**

Accept (Poster)

**Comment:**

Meta Review: The paper incorporates prototype learning in offline policy evaluation for better interpretability.

The reviewers found the idea interesting and useful for interpretability, and the paper is well written. There are concerns that the performance of the method is not competitive enough, so it would be better if the authors could explain a little about this and try to improve the performance, since both performance and interpretability are important. I hope the authors consider the comments of the reviewers when polishing the paper.